# A Brief Overview of Potential Treatments for Viral Diseases Using Natural Plant Compounds: The Case of SARS-Cov

**DOI:** 10.3390/molecules26133868

**Published:** 2021-06-24

**Authors:** Rambod Abiri, Hazandy Abdul-Hamid, Oksana Sytar, Ramin Abiri, Eduardo Bezerra de Almeida, Surender K. Sharma, Victor P. Bulgakov, Randolph R. J. Arroo, Sonia Malik

**Affiliations:** 1Department of Forestry Science and Biodiversity, Faculty of Forestry and Environment, Universiti Putra Malaysia, Serdang 43400, Malaysia; rambod.abiri@gmail.com or; 2Laboratory of Bioresource Management, Institute of Tropical Forestry and Forest Products (INTROP), Universiti Putra Malaysia, Serdang 43400, Malaysia; 3Educational and Scientific Center “Institute of Biology and Medicine”, Department of Plant Biology, Taras Shevchenko National University of Kyiv, Volodymyrska 60, 01033 Kyiv, Ukraine; oksana.sytar@uniag.sk; 4Department of Plant Physiology, Slovak University of Agriculture Nitra, A. Hlinku 2, 94976 Nitra, Slovakia; 5Department of Microbiology, School of Medicine, Kermanshah University of Medical Sciences, Kermanshah 6718773654, Iran; rabiri@kums.ac.ir; 6Fertility and Infertility Research Center, Health Technology Institute, Kermanshah University of Medical Sciences, Kermanshah 6718773654, Iran; 7Biological and Health Sciences Centre, Laboratory of Botanical Studies, Department of Biology, Federal University of Maranhão, São Luís 65080-805, MA, Brazil; eduardo.almeida@ufma.br; 8Department of Physics, Central University of Punjab, Bathinda 151401, India; surender.sharma@cup.edu.in; 9Department of Biotechnology, Federal Scientific Center of the East Asia Terrestrial Biodiversity (Institute of Biology and Soil Science), Far Eastern Branch of the Russian Academy of Sciences, 159 Stoletija Str., 690022 Vladivostok, Russia; 10Leicester School of Pharmacy, De Montfort University, The Gateway, Leicester LE1 9BH, UK; rrjarroo@dmu.ac.uk; 11Health Sciences Graduate Program, Biological & Health Sciences Centre, Federal University of Maranhão, São Luís 65080-805, MA, Brazil; 12Laboratoire de Biologie des Ligneux et des Grandes Cultures (LBLGC), University of Orléans, 1 Rue de Chartres-BP 6759, 45067 Orleans, France

**Keywords:** bioactive compounds, coronavirus, hairy roots, herbal medicines, molecular farming, plant extracts, respiratory diseases

## Abstract

The COVID-19 pandemic, as well as the more general global increase in viral diseases, has led researchers to look to the plant kingdom as a potential source for antiviral compounds. Since ancient times, herbal medicines have been extensively applied in the treatment and prevention of various infectious diseases in different traditional systems. The purpose of this review is to highlight the potential antiviral activity of plant compounds as effective and reliable agents against viral infections, especially by viruses from the coronavirus group. Various antiviral mechanisms shown by crude plant extracts and plant-derived bioactive compounds are discussed. The understanding of the action mechanisms of complex plant extract and isolated plant-derived compounds will help pave the way towards the combat of this life-threatening disease. Further, molecular docking studies, in silico analyses of extracted compounds, and future prospects are included. The in vitro production of antiviral chemical compounds from plants using molecular pharming is also considered. Notably, hairy root cultures represent a promising and sustainable way to obtain a range of biologically active compounds that may be applied in the development of novel antiviral agents.

## 1. Introduction

Bronchitis is a respiratory disease caused by bacterial infections, viral infections, or irritant particles [1]. In response to infection, the bronchial tubes become inflamed and swollen, which may eventually result in acute respiratory arrest. Coronavirus disease-19 (Covid-19), which recently emerged as a pandemic, is an infectious respiratory disease caused by a newly discovered coronavirus. The first report of this novel coronavirus was traced back to the Huanan wholesale seafood market in Wuhan city, China, in December 2019, where a group of patients exhibited a mysterious kind of viral pneumonia [2]. Nowadays, viral pneumonia is diagnosed through analyzing a sample of bronchoalveolar lavage fluid using PCR, cell cultures, and whole-genome sequencing [3]. The virus was isolated from infected individuals and recognized as genus beta-coronavirus, placing it alongside other viruses causing Severe Acute Respiratory Syndrome (SARS) and Middle East Respiratory Syndrome (MERS) [4]. Previously, SARS had been reported in Southern China during 2002–2003, and its outbreak has been reported in 29 countries, with almost 8000 infected cases and around 700 mortalities (http://www.who.int/csr/sars/en/, the accessed date: 7 February 2021). A decade after the appearance of SARS, MERS was reported during 2012–2014 as a second coronavirus generation that caused a global pandemic. MERS affected people in more than 27 countries with over 2000 (32.97%) case-fatality. Coronavirus is an enveloped, positive-sense, and single-stranded RNA or (+) ssRNA virus with a large genome of approximately 30 kb.

Covid-19 is a life-threatening illness with a tremendous rate of spreading in humans due to its high level of infectiousness [5]. The treatment of this disease is a great challenge due to several reasons, including the rapid emergence of mutant strains, the consequent high rate of virus adaptation, and the development of resistance to antiviral medicines. Another factor is that of unwanted side effects and the high cost of synthetic antiviral drugs. There has been a significant global interest in developing safe and effective Covid-19 vaccines since 2020. Forman and his colleagues provided an overview of current phase II/III, III, and IV COVID-19 vaccine candidates (20 different vaccines) [6]. The standard approach for viral infections comprises antiviral medicines that do not cause damage to the human host but can help shorten viral infection, inhibit virus expansion, and help in reducing/blocking complications [1]. In parallel with the generation of vaccines in 2020, studies on the antiviral properties of natural compounds have also been performed via the molecular docking methodology [7,8,9,10]. The potency of marine natural products has been confirmed to target SARS-CoV-2 main protease (Mpro) [11].

Medicinal plants have been identified as reliable resource against several diseases for millennia. More than 70% of the global population still depends on herbal medicines due to their relatively low cost and better compatibility with the human body compared to synthetic drugs [12]. During the pandemic period, studies were performed using databases of scientific literature to screen and identify the potential of herbal plants to act as anti-coronavirus medication [13]. It has been reported that water and ethanol plant extracts contain biologically active substances with antiviral activity [14].

A wide range of compounds identified in several plant species have demonstrated antiviral activities, including alkaloids, flavonoids, triterpenes, anthraquinones, and lignans. Interestingly, plant selection based on ethnomedical concerns provides a higher hit rate than screening plants or general synthetic products [15]. For instance, lycorine extracted from *Lycoris radiata* L. and glycyrrhizin from *Glycyrrhiza uralensis* Fisch showed anti-SARS-CoV activities [16]. Some known pharmacophore structures of bioactive substances may be useful in the creation of new anti-Covid-19 drugs. Natural compounds such as betulinic acid, indigo, aloe emodin, luteolin, quinomethyl triterpenoids, quercetin, and gallates have been reported to inhibit viral proteases with the potential to develop antiviral drugs [14]. In addition, plants have also been introduced as a safe and reliable bioreactor for the production of recombinant virus proteins that can be used in vaccine development [17], e.g., nuclear transformed tomatoes and tobacco-expressing antigens have been reported to induce immunogenic responses against SARS-CoV [18].

The main objective of the current review is to provide the complete overview of the ethnomedicinal uses of herbs employed to treat respiratory diseases. We address questions regarding the potential of plant-derived compounds in inhibiting virus propagation, thus providing relief for viral-induced pathogenesis. We also discuss how biotechnology may help solve the challenge of rapidly obtaining pure antiviral compounds. Furthermore, this review discusses the current state of the art regarding the possible antiviral activities of herbal medicine and makes an effort to tackle the gaps in scientific knowledge that may lead to the advancement of innovative treatments for the welfare of people and against the spread of viral diseases, especially SAR-CoV.

## 2. A Brief Model of Viral Entering/Replication in Host Cells

The life and replication cycle of a virus is dependent on the cell processes of its host (Figure 1). The reproduction cycle of viruses causes significant structural (cytopathic) and biochemical damage that, in severe conditions, may ultimately result in the death of the host organism [19]. Viruses can enter host cells via different biological mechanisms including phagocytosis, pinocytosis, and endocytosis. To enter and spread to the cells of a living organism, the virus fuses at the plasma membrane in the first phase, and then it infects the other cells of the host organism via cell-to-cell syncytia or fusion in the second phase. The respiratory system is the main site of entrance for viruses into host organisms, and severe infection of the respiratory tract may cause life-threatening damage to the lungs [20]. Generally, viral replication is completed within several hours [21] and involves various phases including attachment, penetration, un-coating, replication, assembly, and release [22].

## 3. General Model of SARS-CoV Pathogenicity

Coronaviruses (CoVs) are considered as a subfamily of Orthocoronavirinae and belongs to a cluster of vertebrate animal and human viruses that affect the respiratory tract, liver, central nervous systems, and digestive systems [24]. Currently, no specific anti-viral treatment has been recommended for COVID-19, and investigations have focused on vaccine development [25]. This human pathogen is a positive-strand RNA virus (positive-sense (5′-to-3′) viral RNA) enveloped by spike glycoproteins (S). These proteins bind to their receptor on the surface of the host cells. The spike proteins possess two compartments, namely envelop proteins (E) and membrane proteins (M), which play important roles in pathogenesis. Some spike glycoproteins have envelope-associated hemagglutinin-esterases (HEs) [26]. CoVs can bind to angiotensin-converting enzyme (ACE2) receptors through the receptor-binding domain (RBD) and then enter cells. The virus has two different shapes, L and S. The “L” from is assumed to be more virulent and adaptable than the “S” form to interact with the [27]. Specifically, the S1 domain of Covid-19’s spike glycoprotein interacts with an immunoregulatory factor for virulence and hijacking, the human CD26 [28]. CD26 is linked to inflammation; it plays a vital function in T-cell activation and is expressed in plasma and on the cell surface of several non-immune and immune cell types. It has been reported that high-fiber diets and citrus flavonoids that ameliorate the effects of type 2 diabetes mellitus (T2DM) are also inhibitors of CD26 [20].

Furthermore, the epigenetic dysregulation of ACE2 and interferon-regulated genes may suggest increased Covid-19 susceptibility and severity in lupus patients [29]. The S protein has two functional domains: (a) the receptor-binding domain and (b) sequences that mediate the fusion of viral and cell membranes. S-glycoprotein must be cleaved by cellular proteases to provide access to fusion sequences and is required for cell entry. The nature of the cell protease that cleaves the S glycoprotein varies according to the coronavirus type. For instance, the MERS-CoV S glycoprotein contains a furin protease cleavage site and is processed by these intracellular proteases during exit from the cell. This protein processing by the host’s furin protease is necessary for the development of the virus and allows it to enter the next cell. In contrast, SARS-CoV S glycoprotein is uncleaved upon release of virus from cells. It is probably cleaved during virus entry into a cell. Furin proteases may therefore be targeted for therapeutic uses [20] (Figure 2).

## 4. Replication Inhibitors of SARS-CoV

Previous investigations have demonstrated that the development of proteases is an ideal goal to be tackled for the inhibition of CoV replication. In silico analysis demonstrated a 96% similarity between 2019-CoV main protease and the SARS-CoV M^pro^, and no mutation was reported in the active sites in both proteins [31]. Though the protease activity disruption causes various diseases, host proteases are considered reliable therapeutic targets. For several different viruses, protease activity represents a vital factor in replication; thus, proteases are frequently targeted as protein candidates during antiviral therapeutics studies [32]. Lopinavir and nelfinavir are in the category of medications named protease inhibitors with a high level of cytotoxicity recommended for the treatment of cells infected with MERS, SARS, and HIV [33]. Some preliminary investigations have examined the potential of protease inhibitor ritonavir/lopinavir, which is regularly implemented to cure human immunodeficiency virus (HIV)/acquired immunodeficiency syndrome cases for the therapy of Covid-19-infected patients. In addition, other studies on the antiviral treatment of pathogenic forms of human CoV have reported on lamivudine (3TC), tenofovir disoproxil (TDF), umifenovir (arbidol), remdesivir, neuraminidase inhibitors, and nucleoside analogues, among others [34]. Moreover, another effort demonstrated that nelfinavir was the best potential inhibitor candidate over praziquantel, perampanel, and pitavastatin against COVID-19 Mpro. The efficiency of nelfinavir has been reported due to the binding free energy calculations using the molecular mechanics with generalized Born and surface area solvation (MM/GBSA) model and solvated interaction energy (SIE) methods. In this regard, the crystal structures of protease/chymotrypsin-like protease (3CL^pro^) [27] from Covid-19 patients have shown that this protease could have the capability to inhibit CoV replication [35]. On the other hand, molecular docking investigations revealed that epicatechin-gallate, catechin, curcumin, oleuropein, apigenin-7-glucoside, demethoxycurcumin, and luteolin-7-glucoside may have the potential to inhibit Covid-19 Mpro. Additionally, an in silico analysis showed that the mentioned phenolic compounds share a pharmacophore with nelfinavir [36]. Thus far, different flavonoid compounds derived from medicinal plants have shown antiviral bioactivities for inhibiting protease.

## 5. Inhibitors of Assembly and Packaging of SARS-CoV

During the virus’ life cycle, the assembly and release of infectious particles is the final phase [37]. In this phase, viral structural proteins (often mentioned as pre-structural proteins such as P1 of enterovirus 71) mature until they are assembled into viral capsids. In this phase, the SARS-CoV genomes are actively tightly packaged into pre-formed viral protein capsids for enveloped intracellular cargo transport and then released [38]. Notwithstanding the absolute necessity for virus infection, so far, no antiviral drugs/agents have tackled this phase [39]. However, the results of studies have shown that some medicinal plants can interfere with the viral packaging and assembly mechanisms [23].

## 6. Evidence Supporting the Antiviral Efficacy of Medicinal Plants

The use of therapeutic plants against viral infection can be traced back to the dawn of civilization; however, BOOTS Pure Drug Co., Ltd., Nottingham (England) made the first systematic effort to screen plants against influenza [40]. Later on, the inhibitory effect of medicinal plants on the replication of viruses was studied on severe acute respiratory syndrome (SARS) virus, emerging viral infections linked with poxvirus, hepatitis B virus (HBV), HIV, and herpes simplex virus type 2 (HSV-2) [41,42,43,44,45,46] (Table 1). Many studies have applied either alcoholic or aqueous extracts of medicinal plants. However, only a few investigations have been conducted to study active natural compounds presenting antiviral effects. It has been demonstrated that molecular mechanisms linked to the antiviral effects of medicinal plant extracts vary among various types of viruses. Nonetheless, medicinal plant extracts potentially improve the inherent antiviral defense mechanisms of the human body, which involve a complicated system and might use several concurrent pathways. Thus far, some investigations have discovered immunostimulatory properties of medicinal plant extracts possessing antiviral activity [47]. Many plants including *Vitex negundo, Solanum* nigrum, *Scharicum guerke, Ocimum kilim, Sambucus ebulus, Ocimum sanctum, Euphorbia granulate, Eugenia jambolana,* and *Acacia nilotica* have been revealed to target reverse transcriptase activity and have shown inhibitory actions towards HIV proteases [48,49,50,51,52,53]. The root extracts of *Heracleum maximum* Bartr. (Apiaceae) enhanced interleukin 6 (IL-6) in a macrophage activation test, thus proving antiviral effects linked with the immunostimulatory characteristics [47]. Likewise, *Plantago asiatica* Linn. (Plantaginaceae) and *Plantago* major Linn. are frequently used as folk medicinal plants in Taiwan for the treatment of different infectious diseases, and both were found to induce the secretion of interferon-gamma (IFN-γ) and lymphocyte proliferation at low concentrations. Both the secretion of interferon-gamma (IFN-γ) and the induction of lymphocyte proliferation activity are reliable indicators of cell-mediated immune response modulation [47,54].

## 7. Plant-Derived Immunomodulators

The phagocyte–microbe interactions in the immune system comprise a defense reaction that, under more harmful circumstances, may take part in the advancement of various immune and non-immune chronic inflammatory diseases. The immune system of a healthy organism manages the homeostasis of the body. Agents that express a capacity to modulate and normalize pathophysiological processes are named immunomodulators [116]. Most of the well-known immunostimulants and immunosuppressants used in clinical practice are cytotoxic drugs, which can have severe side effects. Therefore, plant-derived compounds and extracts have been studied regarding their immunomodulatory potential in humans due to their lower cytotoxicity and high bioavailability [117,118]. Plant-derived immunomodulators can also be used for a long period [119] (Table 2).

Some plant-derived compounds, e.g., curcumin, genistein, fisetin, quercetin, resveratrol, epigallocatechin-3-gallate, andrographolide, and colchicine, have immunomodulatory effects [120,121,122,123,124,125,126]. These compounds can downregulate the production of proinflammatory cytokines induced by some viroidal agents [122]. Andrographolide and other natural immunomodulators can enhance the activity of cytotoxic T cells, phagocytosis, natural killer (NK) cells, and antibody-dependent cell-mediated cytotoxicity [125]. The use of quercetin in combination with highly active compounds such as psoralen, baccatin III, embelin, and menisdaurin increased its anti-hepatitis B activity up to 10% [127].

At the same time, analyses of other medicinal plant extracts with antiviral properties have shown that *Gymnema sylvestre* [121], *Stephania tetrandra* S Moore roots [128,129,130], and *Vitex trifolia* extracts [107] possess immunomodulating activity. Naser et al. [131] found immunomodulation potential and antiviral activities against acute common cold in leaf extracts of *Thuja occidentalis*. The anti-SARS and immunomodulatory activity of water extracts of *Houttuynia cordata* have been reported via the stimulation of lymphocyte proliferation together with enhancing the proportion of CD^4+^ and CD8^+T^ cells [81] (Table 3).

**Table 2 molecules-26-03868-t002:** Plant sources of polyphenolic compounds with anti-protease activity.

Plant Species	Sources	Compounds	Molecular Formula	Lipinski’s Rule of Five		Reference
				Properties	Value	
*Spinacia oleracea,* *Brassica oleracea, Anethum graveolens, Brassica rapa, Sauropus androgynus*	Spinach Cabbage Dill Chinese cabbage Katuk	Kaempferol	C15H10O6	Molecular weight (<500 Da)	286.24	[132,133]
				LogP (<5)	1.58	
				H-bond donor (5)	4	
				H-bond acceptor (<10)	6	
				Violations	0	
*Anethum graveolens, Foeniculum vulgare, Allium cepa,* *Oregano vulgare, Capsicum annum*	Dill Fennel leaves Onion Oregano Chili pepper	Quercetin	C15H10O7	Molecular weight (<500 Da)	302.24	[132]
				LogP (<5)	1.23	
				H-bond donor (5)	5	
				H-bond acceptor (<10)	7	
				Violations	0	
*Olea europaea, Averrhoa belimbi, Capsicum annum, Allium fistulosum*	Olive Star fruit Chili pepper Welsh onion/Leek	Luteolin-7-glucoside	C21H20O11	Molecular weight (<500 Da)	448.38	[134]
				LogP (<5)	0.16	
				H-bond donor (5)	7	
				H-bond acceptor (<10)	11	
				Violations	2	
*Curcuma longa*, *Curcuma xanthorriza*	Turmeric Curcuma	Demethoxycurcumine	C20H18O5	Molecular weight (<500 Da)	338.35	[135,136]
				LogP (<5)	3	
				H-bond donor (5)	2	
				H-bond acceptor (<10)	5	
				Violations	0	
*Citrus sinensis*	Citrus fruit	Naringenin	C15H12O5	Molecular weight (<500 Da)	567.78	[137]
				LogP (<5)	4.33	
				H-bond donor (5)	4	
				H-bond acceptor (<10)	5	
				Violations	1	
*Averrhoa belimbi, Lycium chinese, Apium graveolens, Olea Europaea*	Star fruit Goji berries Celery Olive	Apigenine-7-glucoside	C21H20O10	Molecular weight (<500 Da)	432.34	[138,139,140]
				LogP (<5)	0.55	
				H-bond donor (5)	6	
				H-bond acceptor (<10)	10	
				Violations	1	
*Olea Europaea*	Olive	Oleuropein	C19H22O8	Molecular weight (<500 Da)	378.37	[138]
				LogP (<5)	1.57	
				H-bond donor (5)	3	
				H-bond acceptor (<10)	8	
				Violations	0	
*Camellia sinesis*	Green tea	Catechin	C15H14O6	Molecular weight (<500 Da)	290.27	[141,142]
				LogP (<5)	0.85	
				H-bond donor (5)	5	
				H-bond acceptor (<10)	6	
				Violations	0	
*Curcuma xanthorriza, Curcuma longa*	Turmeric Curcuma	Curcumin	C21H20O6	Molecular weight (<500 Da)	368.38	[135,136]
				LogP (<5)	3.03	
				H-bond donor (5)	2	
				H-bond acceptor (<10)	6	
				Violations	0	
*Zingiber officiale*	Ginger	Zingerol	C11H16O3	Molecular weight (<500 Da)	196.24	[36,143,144]
				LogP (<5)	1.86	
				H-bond donor (5)	2	
				H-bond acceptor (<10)	3	
				Violations	0	
*Zingiber officiale*	Ginger	Gingerol	C17H26O4	Molecular weight (<500 Da)	294.39	[36,144,145]
				LogP (<5)	3.13	
				H-bond donor (5)	2	
				H-bond acceptor (<10)	4	
				Violations	0	
*Allium sativum*	Garlic	Allicin	C6H10OS2	Molecular weight (<500 Da)	162.27	[36]
				LogP (<5)	1.61	
				H-bond donor (5)	0	
				H-bond acceptor (<10)	1	
				Violations	0	
*Camellia sinesis*	Green tea	Epicatechin gallate	C22H18O10	Molecular weight (<500 Da)	442.37	[139]
				LogP (<5)	1.23	
				H-bond donor (5)	7	
				H-bond acceptor (<10)	10	
				Violations	1	

**Table 3 molecules-26-03868-t003:** The mode of action against viruses and methods of active compound extraction from medicinal plants.

Plant Species and Plant Part	Active Compounds				Coumarins	Extract	Model Organism	Mode of Action/Activity	Ref
	Terpenes Terpenoids	Flavonoids Flavones	Alkaloids	Stilbenes					
*Méntha piperíta* (whole plant) *Lamiaceae*	α-pinene β-pinene β-caryophyllene L-Limonene Menthol	Eriocitrin Hesperidin Kaempferol 7-*O*-rutinoside Luteolin and its derivatives	n/a	Trans-resveratrol	n/a	Ethanol	Vero cell cultures	High antiviral activity	[145,146,147]
*Thymus vulgaris* (whole plant) *Lamiaceae*	Thymol p-cymene g-erpinene γ-Terpinene Linalool	Rutin Quercetin	n/a	n/a	n/a	Ethanol	Vero cell cultures	High antiviral activity and antioxidant effects	[145,148,149]
*Desmodium canadense* (whole plant) *Fabaceae*	Sandosaponin B and its derivativesSoyasaponin I Soyasaponin VI	Homoorientin Orientin 2-vicenin Vitexin Isovitexin Rutin Desmodin Homoadonivernite	Indole-3-alkylamine phenylethylamine alkaloids, pyrrolidine alkaloids	n/a	n/a	Ethanol	Vero cell cultures	High antiviral activity	[145,150,151,152,153]
*Camellia japonica* (whole plant, flowers) *Theaceae*	Oleanane triterpenes 3β,18β-dihydroxy-28-norolean-12-en-16-one 18β-hydroxy-28-norolean-12-ene-3,16-dione	Quercetin Kaempferol Apigenin	Do not produce purine alkaloids	n/a	n/a	Ethanol	Vero cells (African green monkey kidney cell line; ATCC CCR-81)	High antiviral activity on PEDV corona virus Inhibitory effects on key gene and protein synthesis during PEDV replication	[154,155,156,157,158,159]
*Saposhnikovia divaricate* (whole plant) *Apiaceae*	n/a	n/a	n/a	n/a	cis-3′-Isovaleryl4′-acetylkhellactone Praeruptorin F Praeruptorin B (−)-cis-khellactone	Ethanol	Vero cells (African green monkey kidney cell line; ATCC CCR-81)	High antiviral activity on PEDV corona virus	[155]
*Quercus ilex* L. (Leaves) *Fagaceae*		kaempferol glycosides (juglanin, kaempferol-3-*O*-*α*-L-arabinofuranoside, and afzelin, kaempferol-3-*O*-*α*-L-rhamnoside	n/a	n/a	n/a	DMSO	*Xenopus oocytes*	Inhibits 3a channel protein of coronavirus	[156,157]
*Bupleurum sp.* (whole plant) *Apiaceae*	Triterpenoid saponins Saikosaponins 2″-*O*-Acetylsaikosaponins Prosaikogenins Bupleurosides Etc.	Quercetin Isorhamnetin Narcissin Rutin Eugenin Saikochrome A	n/a	n/a	n/a	DMSO	Human fetal lung fibroblasts (MRC-5; ATCC CCL-171)	Saikosaponins attenuate viral attachment and penetration	[160,161]
*Houttuynia cordata* (whole plant) (*Saururaceae*)	Cycloart-25-ene-3b,24-diol	Quercetin 7-rhamnoside Hyperin Quercetin Afzelin Rutin	Arisolactams Piperolactam A Caldensin	n/a	n/a	Water	BALB/c mice	Decreases the viral SARS-3CL^pro^ activity Stops viral t RNA polymerase activity (RdRp) Increases the secretion of interleukin (IL)-2 and (IL)-10	[88,162]
*Isatis tinctoria* (Roots extracts) *Brasicaceae*	n/a	Hesperetin Quercetin Isoorientin Isovitexin	Indigo Indirubin Indican	Sinigrin	n/a	Water	Vero cells	Cleavage of the activity of SARS-3CL^pro^ enzyme decreased	[163,164]
*Lycoris radiata* (Bulbs) *Amaryllidaceae*	β-Myrcene A-terpineol Eucalyptol β-cyclocitral	n/a	Lycorine Amaryllidaceae alkaloids Lycoranines	n/a		Ethanol	Vero E6 cells	Exhibits anti-SARS-CoV activity	[16,165,166,167]
*Litchi chinensis* (seeds) *Sapindaceae*	3-Oxotrirucalla-7,24-dien-21-oic acid	Herbacetin Rhoifolin Pectolinarin Quercetin Epigallocatechin gallate Gallocatechin gallate Litchitannins Kaemferol derivatives Epicatechin Cinnamtannin	n/a	n/a	n/a	Water	On model with SARS-CoV 3CL^pro^	Inhibits SARS-3CL^pro^ activity	[168,169,170,171]
*Stephania tetrandra* S Moore (Roots) *Menispermaceae*	n/a	n/a	Tetrandrine Fangchinoline, Cepharanthine	n/a	n/a	DMSO	Human cell line MRC-5 cells	Inhibits the expression of HCoV-OC43 spike and nucleocapsid protein. Immunomodulation/	[129,172]
*Scutellaria baicalensis* (Roots) *Lamiaceae*	Dodecanedioxins	Scutellarein Baicalin Wogonin Wogonoside	n/a	n/a	n/a	DMSO	Model with SARS-CoV helicase, and nsP13	Inhibits nsP13 by affecting the ATPase activity	[61,173]
*Allium sativum* (Bulbs) *Alliaceae*	Nerolidol Phytol Squalene α-pinene Terpinolene Limonene 1,8-cineole γ-terpinene	Catechin Epicatechin	*Allicin* Ajoene Alliin Diallyl disulfide Diallyl trisulfide	n/a	n/a	Aquaporin	Chicken embryos	Inhibitory effects on avian coronavirus	[174,175,176,177]
*Artemisia sp.* *Artemisia absentium* (whole plants) *Asteraceae*	Absinthin Artemisin Scopoletin Artamarin	Rutin Glycosides of quercetin	*Artamarin* Artamaridin, Artamaridinin, Artamarinin QuebrachitolArtemitin	n/a	n/a	Water	Delayed brain tumor cells	Reduces coronavirus replication	[178,179]
*Juniperus communis* (Fruits) *Cupressaceae*	Sugiol *α*-pinene *β*-pinene	Rutin Scutellarein Quercetin-3-*O*-rhamnoside quercitrin	n/a	n/a	Umbelliferone	n/a	Protein-molecular docking with network pharmacology analysis	Inhibits the replication, 3CL^pro^	[180,181]
*Ecklonia cava* (whole plant) *Lessoniaceae*	n/a	Quercetin	n/a	n/a	n/a	n/a	protein-molecular docking with network pharmacology analysis	PL^pro^ and 3CL^pro^	[182]

### 7.1. Lectins

Lectins are a special type of natural proteins (split into seven different classes of evolutionarily- and structurally-related proteins) found in higher plants that bind to the sugar moieties of a wide range of glycoproteins [180]. Plant lectins can inhibit virus replication by preventing the adsorption and fusion of HIV in lymphocyte cell cultures [181,182,183,184,185,186,187,188,189]. Furthermore, the antiviral effect of agglutinins specific for N-acetylglucosamine and mannose on HIV has been reported. The inhibitory effect of these plant lectins has been shown in vitro on infection with influenza A virus, respiratory syncytial virus, and cytomegalovirus [188,189,190]. The SARS-CoV spike protein contains 23 putative N-glycosylation sites and is heavily glycosylated. Among the putative N-glycosylation sites, 12 have been defined to be glycosylated [191]. It can be expected that the infectivity of the coronavirus will be suppressed by those lectins that are specific to the glycans present in the spike glycoprotein. The antiviral effect of mannose-specific plant lectins has been reported against coronavirus.

### 7.2. Quercetin

Quercetin is a plant-derived flavonol (pigment) that is commonly found in vegetables and fruits. It is known to possess an antioxidant, antiviral, anti-inflammatory, and anti-carcinogenic effects, which may reduce risk of infection and improve physical or mental performance. Moreover, quercetin facilitates the ability to stimulate mitochondrial biogenesis and inhibit capillary permeability, platelet aggregation, and lipid peroxidation [20]. Quercetin is widely found in leaves, flowers, barks, nuts, and seeds of a variety of plants such as tomatoes, tea, shallots, grapes, capers, *Brassica* vegetables, berries, apples, *Sambucus canadensis*, *Hypericum perforatum*, and *Ginkgo biloba* [192]. The recommended consumption of this plant-derived flavonol has been stated to be not less than 4.37 mg/day. Though the highest concentration of quercetin is reported in capers (234 mg of flavonol per 100 g of edible portion), the main plant source for quercetin glycosides is apples, with 7.4% 13 mg/100 g fruit. During the digestion of food, quercetin and its conjugated metabolites can be converted into a range of metabolites (phenolic acids) by enteric enzymes and bacteria in intestinal mucosal epithelial cells (IMECs) [191]. Quercetin also inhibits the senescence-associated pro-inflammatory response and suppresses stress-induced senescent cells [193]. Numerous in vitro investigations have confirmed the inhibitory impact of quercetin on interleukin 8 (IL-8) and *tumor*-necrosis factor (TNF-α) production in cells. Additionally, several studies have shown the protective function of this flavonol against inflammation in human umbilical vein endothelial cells (HUVECs), as well as mediation via the downregulation of vascular cell adhesion molecule 1 (VCAM-1) and CD80 expression [20,193]. Quercetin considerably induces the production of derived interferon (IFN) and T helper type 1 (Th-1), and it consequently downregulates Th-2-derived interleukin 4 (IL-4) by normal peripheral blood mononuclear cells. Quercetin has been found to suppress the infection caused by a wide spectrum of influenza strains including A/Puerto Rico/8/34 (H1N1), A/FM-1/47/1 (H1N1), and A/Aichi/2/68 (H3N2) with half-maximal inhibitory concentrations (IC_50_) of 7.76, 6.22, and 2.74 μg/mL, respectively [20,194,195]. Investigations into influenza mechanisms have shown the positive interactions between the viral HA2 subunit (a mark for antiviral vaccines) and quercetin. This discovery may determine the antiviral potential of quercetin in the early stages of influenza. Furthermore, this anti-viral compound could prevent H5N1 virus entry into the cell [20,194,196].

### 7.3. Sulforaphane

Sulforaphane is an isothiocyanate (isothiocyanate sulforaphane (SFN)) that has been stated to be an antiviral agent. It has been reported that the osteoblast supporting transcription factor Runx2 is essential for the long-term perseverance of antiviral CD^8+^ memory T cells [197,198]. An addition, SFN-rich broccoli homogenate attenuated granzyme B production in NK cells that was induced by influenza virus and granzyme B production in NK cells, and granzyme B levels appeared to have negatively interacted with influenza RNA levels in nasal lavage fluid cells [199]. Nasal influenza infection can induce complex cascades of changes in peripheral blood NK cell activation. SFN increases as a result of virus-induced peripheral blood NK cell granzyme B production, which may enhance antiviral defense mechanisms [20,199].

### 7.4. Resveratrol

Resveratrol is a natural polyphenol found in grapes, mulberry, and peanuts. It is known to have antiviral properties against a variety of viral pathogens in vitro and in vivo [200]. Resveratrol is available in *trans*- and *cis*-isomeric forms. The *cis*-resveratrol isomer is unstable and can be easily transformed into the trans form when it reacts with light. It was demonstrated that resveratrol substantially inhibited MERS-CoV replication in vitro through the inhibition of RNA production, as well as other pleiotropic effects. Studies have shown that indomethacin and resveratrol can act as adjuncts for SARS-CoV-2/COVID-19 [201,202]. Medina-Bolivar et al. developed the hairy root lines from *Arachis hypogaea* (peanut) for the sustained and reproducible production of resveratrol and resveratrol derivatives [201].

### 7.5. Baicalin

Baicalin (baicalein glucuronide) accumulates in the roots of *Scutellaria baicalensis* [202]. Baicalin has been reported as an antioxidant possessing anti-apoptotic properties, and it has been used for pulmonary arterial hypertension treatment [203]. This flavone glucuronide has been reported to have anti-SARS-CoV inhibitory effects comparable to those of interferon-beta 1a, interferon-alpha, and glycyrrhizin. At the same time, this compound has a low toxicity in human cell lines [204]. Baicalin showed considerable anti-viral properties on lipopolysaccharide-activated cells, while the oral application of baicalin expressively increased the survival rate of influenza A virus-infected mice [57]. The in silico analysis of the inhibitory effect of baicalin showed that this flavone inhibits ACE2 in the case of COVID-19 disease. It has been revealed that baicalin can inhibit 3CL^Pro^ activity of the SARS-Cov2 virus in vitro [205].

### 7.6. Glycyrrhizin

Glycyrrhizin (a triterpene saponin) is one of the most important phytochemical components of the *Glycyrrhiza glabra* (licorice) root [206]. Glycyrrhizin has anti-inflammatory and antioxidant properties used for treatments of different diseases such as jaundice, bronchitis, and gastritis [204]. Glycyrrhizin could block the SARS-Cov virus attachment to the host cells, especially during the initial stage of the viral life cycle [207]. An in silico analysis of glycyrrhizin behavior showed the inhibitory effect of this compound on SARS-Cov2 [208].

### 7.7. Narcissoside

Narcissoside (synonym: narcissin) is a phytochemical belonging to the group of mono-methoxyflavones. This isorhamnetin-3-*O*-rutinoside flavonoid is extracted from leaves of various folk plants such as *Atriplex halimus* L., *Gynura divaricate*, *Caragana spinose*, and *Manihot escylenta*. An in silico analysis demonstrated that narcissoside has inhibitory potential for the viral COVID 19 protein 6W63 [209].

### 7.8. Curcumin

Curcumin is diarylheptanoid that possesses inflammatory and antioxidant properties and is mainly extracted from *Alpinia galanga*, *Curcuma longa*, and *Caesalpinia sappan* [210]. This active compound has been applied in the treatment of hyperlipidemia, anxiety, arthritis, and metabolic syndrome [211]. Additionally, curcumin displays antibacterial and antiviral properties against *Pseudomonas*, *Streptococcus*, and *Staphylococcus* strains, as well as HIV, hepatitis C, and the influenza virus. Moreover, antiviral properties of this compound have been reported against chikungunya virus (CHIKV), human papillomavirus (HPV), HIV-1 and HIV-2 proteases, emerging arboviruses like the Zika virus (ZIKV), influenza viruses, HIV, HSV-2, and hepatitis viruses [212]. However, due to its rapid elimination, rapid metabolism, and poor absorption, curcumin has poor bioavailability, which reduces its therapeutic effect [213]. It has been reported that the combination of this diarylheptanoid with other chemical compounds like piperine can increase bioavailability (by up to 2000%) and provide multiple benefits to human health [214,215]. This compound can diminish various forms of free radicals, such as reactive nitrogen and oxygen species, and modulate the function of SOD [216], catalase, and GSH enzymes in the neutralization of free radicals [217]. Furthermore, curcumin can block ROS-generating enzyme activities such as xanthine oxidase/hydrogenase and cyclooxygenase/lipoxygenase [20]. It has been reported that this plant-derived compound can inhibit the NF-κB activation caused by numerous inflammatory stimuli such as markers of soluble vascular cell adhesion molecule 1 (sVCAM-1), IL-1 beta, IL-6, and inflammation (soluble CD40 ligand (sCD40L)). The results of studies have shown that curcumin can inhibit SARS-CoV through binding to three different protein receptors: SARS-CoV-2 protease (PDB ID:6LU7), PD-ACE2 (PDB ID: 6VW1), and RBD-S (PDB ID:6LXT) [218].

### 7.9. Epigallocatechin Gallate

Epigallocatechin gallate or epigallocatechin-3-gallate is the ester of gallic acid and epigallocatechin [219]. This compound possesses activity against neurological diseases, premature aging, metabolic diseases, and inflammation [220]. Epigallocatechin gallate has shown anti-inflammatory and antioxidative properties, facilitated DNA repair and stability, ensured the modifications of miRNAs, and modulated the epigenetic methylation of histones [20,221]. The antiviral activity of this compound has been reported against a broad spectrum of viruses such as hepatitis B virus (HBV; Hepadnaviridae), human papillomavirus (HPV; Papovaviridae), adenovirus (Adenoviridae), and herpes simplex virus (HSV; Herpesviridae). It has been observed that epigallocatechin gallate can inhibit (+)-RNA viruses such as chikungunya virus (CHIKV; Togaviridae), West Nile viruses (WNV; Flaviviridae), dengue virus (DENV; Flaviviridae), Zika virus (ZIKV; Flaviviridae), and hepatitis C virus (HCV; Flaviviridae). On the other hand, it can inhibit (−)-RNA viruses such as influenza virus (Orthomyxoviridae), Ebola virus (EBOV; Filoviridae), and HIV (Retroviridae) [13]. Epigallocatechin gallate inhibits the early stage of infections, such as attachment, entry, and membrane fusion, by interfering with viral membrane proteins [222]. The anti-viral mechanism of this compound may be generated from interactions with helicase, ACE-2, and DNMTs [20].

## 8. Laboratory Evidence Supporting Application of Medicinal Plants Against Respiratory Disorders

Some investigations have identified the antiviral properties of medicinal plants against bronchitis. For example, it has been demonstrated that *Verbascum thapsus, Justicia adhatoda*, and *Hyoscyamus niger* can reduce infections risks associated with influenza viruses [108]. Polyphenol, extracted from *Cistus incanus* (a Mediterranean plant), possesses anti-influenza activity in MDCK and A549 cell cultures infected with human influenza strains and different subtypes of the avian virus [223]. Similarly, sambucol, extracted from *Sambucus nigra*, showed positive activity against influenza by boosting immune responses through the secretion of inflammatory cytokines (IL-1 beta, TNF-alpha, IL-6, and IL-8) [224].

### 8.1. Artemisia annua

*Artemisia annua* (commonly known as sweet wormwood) is a traditional Chinese medicinal herb. It is a source of the sesquiterpene lactone artemisinin, which is used for the manufacturing of the antimalarial drugs artemether and artesunate [225]. The results of an investigation demonstrated that artemisinin is a reliable source to act as an antiviral compound [226]. Furthermore, sterols extracted from *A. annua* presented inhibitory potential against viruses [227]. An in vitro investigation on the antiviral properties of *A. annua* against SARS-CoV showed positive feedback by using an ethanolic extract with a 50% effective concentration (EC_50_) value of 34.5 ± 2.6 μg/mL and 5a 0% cytotoxic concentration (CC_50_) of 1053 ± 92.8 μg/mL. This result suggested the possibility of applying *A. annua* against SARS-CoV infectious diseases [16]. An analysis of antiviral activity of methanolic extracts of *A. annua* against herpes simplex virus type 1 showed the antiviral properties of the aerial parts of plant. It has been suggested that high levels of bioactive compounds are the main reasons for antiviral activities, which has led to the use of this plant as a potential candidate against viruses [226]. Pulmonary fibrosis (lungs become scarred) is caused by the infection of SARS-CoV with spiked severity, which mediated by interleukin-1 [227]. It has been shown that the consumption of natural antioxidants like polyphenolic compounds [228] is effective in the treatment of lung fibrosis associated with oxidative stress [229]. The positive effect of artesunate, an artemisinin-based drug, to treat pulmonary fibrosis was confirmed due to it inhibiting pro-fibrotic molecules linked to pulmonary fibrosis [230].

### 8.2. Allium cepa

*Allium cepa* (commonly known as onion) is a common raw material served in salad and is rich in natural resources of *organosulfur* and flavonoids with antioxidant activity [231]. Reportedly, quercetin and isorhamnetin (two potential therapeutic agents in onion) can reduce blood pressure and prevent angiotensin-II-induced endothelial dysfunction. Furthermore, superoxide production was increased and subsequently and led to a high nitric oxide bioavailability [232]. It has been reported that the chemical compounds of onion-like flavonoids or even prolines possess antiviral properties against respiratory viruses by stimulating proinflammatory cytokines [233].

### 8.3. Andrographis paniculata

*Andrographis paniculata* (which is commonly known as King of Bitters) is extensively applied in the treatment of several ailments like liver disorders, viral fever, cold, and cough [231]. *Andrographis paniculata* has shown a strong therapeutic effect against viral respiratory infections [234,235,236,237] by suppressing interleukin-1β molecules and increasing NOD-like receptor protein 3 (NLRP3) and caspase-1, which are widely known to play roles in SARS-CoV and likely SARS-CoV-2 pathogenesis [238,239]. Andrographolide (a diterpenoid), is the main bioactive compound extracted from the leaves and stem of this plant and possesses anti-inflammatory capacity. Notably, andrographolide has antiviral potential against different viruses diseases such as chikungunya virus, human immunodeficiency virus, human papillomavirus, Epstein-Barr virus, herpes simplex virus, hepatitis C and B, and influenza A virus (H1N1, H5N1, and H9N2) [125].

### 8.4. Aloe vera

*Aloe vera* is a kind of medicinal plant that possesses antiviral activity against different viruses including human papillomavirus, cytomegalovirus, poliovirus, influenza virus, human immunodeficiency virus, varicella-zoster virus, herpes simplex virus type 2, herpes simplex virus type 1, and hemorrhagic viral rhabdovirus septicemia. Molecular investigations have shown the effectiveness of this plant against other viruses by different action mechanisms such as the breakdown of the viral envelope and interactions with virus enzymes. The presence of some minerals like zinc, copper (Cu), iron (Fe), potassium (K), sodium (Na), magnesium (Mg), and calcium (Ca) make *Aloe vera* a suitable candidate against SARS-CoV-1 [240]. Reports have shown that Zn^2+^ blocks arterivirus RNA polymerase and SARS-CoV activity and inhibits SARS-CoV replication in cell lines [241] (Figure 3).

### 8.5. Nigella sativa

*Nigella sativa* (black seed) is a kind of medicinal plant used for the treatment of a variety of diseases, disorders, and conditions pertaining to the respiratory system, immune systems and cardiovascular, liver, kidneys, and digestive tract. Recently, the result of studies showed that the oil extract of N. sativa reduced the H9N2 avian influenza virus (which is fundamentally associated to SARS-CoV-2 pathogenicity in chicken), and consequently supported the immune response. The most important active compounds which have been identified in *N. sativa* are monoterpenes, e.g., t-anethol, 4-terpineol, carvacrol, p-cymene, thymohydroquinone, and thymoquinone or dimers thereof like dithymoquinone [20].

### 8.6. Salvia officinalis

*Salvia officinalis* (sage) is a medicinal plant with antiviral, antibacterial, antimalarial and antifungal effects. Reportedly, the antiviral and antifungal properties of this plant are most probably facilitated by sageone, safficinolide, and diterpenoids [242].

### 8.7. Toona sinensis

The crude oil of the tender leaves of *Toona sinensis* Roem induced apoptosis in A549 lung cancer cells but also improved the lipolysis of differentiated 3T3-L1 adipocytes [243,244]. Further, the leaf extract of *T. sinensis* Roem was reported to relieve hyperglycemia via modifying adipose glucose transporter 4 [245]. Studies of purified compounds of *T. sinensis* leaves have shown several compounds include toosendanin, phytol, stigmasterol glucoside, beta-sitosterol-glucoside, stigmasterol, beta-sitosterol, (−)-epicatechin, (+)-catechin, kaempferol-D-glucoside, rutin, quercitrin, quercetin, kaempferol, gallic acid, and methyl gallate [246]. Additionally, it has been reported that, quercetin, one of *T. sinensis* leaves’ compounds, has an antiviral impact against HIV-luc/SARS pseudotyped virus [65].

### 8.8. Eckolina cava

*Eckolina cava* (*Laminariaceae*) is a type of brown alga with anti-viral activity against influenza virus neuraminidase and HIV-1 reverse transcriptase 11 phlorotannin chemotype (a diphenyl ether-linked dieckol) extracted from *E. cava* (IC_50s_ = 2.7 and 68.1 µM) that was found to highly block the cleavage of SARS-CoV 3CL^pro^ in a cell-based test with no toxicity [182].

### 8.9. Isatis indigotica

*Isatis indigotica* (*I. indigotica* root *Radix isatidis*) is a Chinese medicinal plant belonging to the family of Cruciferae, with a high phenolic content in the root. The root of this plant was frequently applied during the outbreak of SARS in Taiwan, Hong Kong, and China. Additionally, the antiviral effects of different compounds of this plant such as sinigrin, γ-sitosterol, indican (indoxyl-β-D-glucoside), β-sitosterol, indirubin, and indigo have been reported against Japanese encephalitis, hepatitis A, and influenza [247,248]. Indirubin and indigo were recognized as inhibitors of promiscuous chymotrypsin [249]. Moreover, the antiviral effects of naringenin, quercetin, hesperetin, phenolics, and aloe emodin derived from *I. indigotica* have been accredited against parainfluenza virus, sindbis virus, herpes simplex virus types 1 and 2, vesicular stomatitis virus, poliovirus, and vaccinia virus [250,251]. It has been well-demonstrated that the 3C-like protease (3CL^pro^) can mediate the proteolytic processing of polypeptides 1a and 1ab into functional proteins, which is an ideal target for the development of drugs against SARS-coronavirus. In a cell-based assay, the investigation of seven different phenolic compounds derived from *I. indigotica* revealed that only two polyphenols, namely hesperetin (8.3 µM) and emodin (366 µM), could inhibit the cleavage activity of the 3CL^pro^ in a dose-dependent manner [163].

### 8.10. Azadirachta indica

*Azadirachta indica* (commonly known as neem) is a kind of biological antiviral agent against duck plague virus, herpes simplex virus type-1, bovine herpesvirus type-1 (BoHV-1), poliovirus type 1, group B coxsackieviruses, polio, and dengue virus type-2, as well as infectious bursal diseases like viral infections, Newcastle disease, and highly pathogenic avian influenza virus (H5N1) [252]. It has been shown that different extracts of this neem’s explants have potential against common clinical symptoms of Covid-19 [253,254]. In this regard, crude leaf extracts of neem could be effective against malarial and normal fever [246], as well as a gastrointestinal disorders [255]. The leaf extract of this plant possesses strong antioxidant potential by directly scavenging the hydroxyl radical and preventing hydroxyl radical-mediated oxidative damage in the rat model [256].

### 8.11. Other Medicinal Herbs

Studies on some medicinal plants including *Evolvulus alsinoides*, *Pergularia daemi*, *Clerodendrum inerme Gaertn*, *Clitoria ternatea*, *Sphaeranthus indicus*, *Cassia alata*, *Leucas aspera*, *Abutilon indicum*, *Gymnema sylvestre*, *Vitex trifolia*, and *Indigofera tinctoria* (AO) have shown anti-mouse coronaviral activity [107]. *C. inerme Gaertn* was observed as a promising medicinal plant having potential to inactivate the viral ribosome [104]. Additionally, anti-inflammatory cytokines are significantly reduced by *Sphaeranthus indicus* and *Vitex trifolia* when using the nuclear factor kappa-light-chain-enhancer of activated B (NF-kB) signaling cascade, associated with acute respiratory distress syndrome [152] in SARS-CoV [256,257]. Furthermore, *C. ternatea* (Asian pigeonwings) has been reported as an inhibitor of metallopeptidase domain 17 (ADAM17), a metalloproteinase involved in angiotensin being converted for enzyme shredding, and can be targeted with *Clitoria ternatea*. ACE-2 shredding is associated with the spike formation of viruses [105]. Likewise, *Strobilanthes cusia* was found to inhibit viral RNA genome synthesis and to induce papain like protease (PL^pro^) activity targeting the human coronavirus OC43 [1,113]. *Allium sativum* and *Glycyrrhiza glabra* have been identified to inhibit the viral replication of SARS-CoV. Inhibitory effects on the Ca^2+^ channel were observed when *Hyoscyamus niger* was applied as a bronchodilator [108]. This plant is able to target the orf3a Ca^2+^ channels that attack downstream pathways upon viral infection. The inhibitory effect of other medicinal plants including *Embelia ribes*, *Cassia occidentalis*, *Punica granatum*, *Coscinium fenestratum*, *Cynara scolymus*, *Boerhaavia diffusa*, and *Coriandrum sativum* have been identified against ACE [107]. However, Punica granatum has been shown to exhibit a competitive style of action against virus infection [102,103]. *Salacia oblonga* was found to exhibit suppressive impacts on angiotensin II, an AT1 signal, which was related to lung damage [109] (Table 4).

## 9. In Silico Analysis of Medicinal Plants Role Against SARS-CoV

The in silico analysis of 18 extracted compounds of 11 Indian herbal plants demonstrated different inhibitory properties against Covid-19. Based on the data achieved through log S and log P, as well as the binding affinity (Table 5), the potential inhibitory effect of plants were in the following order: *Nyctanthes arbortristis* (harsingar) > *Aloe barbadensis* miller (aloe vera) > *Tinospora cordifolia* (giloy) > *Curcuma longa* (turmeric) > *Azadirachta indica* (neem) > *Withania somnifera* (ashwagandha) > *Allium cepa* (red onion) > *Ocimum sanctum* (tulsi) > *Cannabis sativa* (cannabis) > *Piper nigrum* (black pepper). However, the results of this investigation confirmed that harsingar, *Aloe vera*, and giloy are more reliable natural resources for future investigations [305]. The in silico analysis of compounds with anti-viral, anti-malaria, or other similar activities are presented in Table 5.

On the other hand, an in silico molecular docking study on thirty-six phytochemical compounds showed high binding affinities for betulinic acid (−10.0 Kcal/mol), silibinin (−9.13 Kcal/mol), oleanolic acid (−9.08 Kcal/mol), and epigallocatechin-3-gallate (−8.51 Kcal/mol). These results suggested that the medicinal plants containing the mentioned compounds are potential candidates against Covid-19 [306].

## 10. Biotechnological Production of Vaccines

At time of writing of this document, no antiviral medicines have been proven to be effective against Covid-19 [307]. As mentioned earlier, remdesivir and hydroxychloroquine have been documented as the most reliable candidates against SARS-CoV [308]. On the other hand, a lot of efforts have been made to develop monoclonal antibodies and to evaluate receptor blockers [309]. Additionally, the use of transfusions by *employing* the plasma of recovered donor from the SARS-CoV virus infection is under assessment [310]. Over the last few decades, therapeutic vaccines have been revealed as the most effective tool against infectious diseases [311]. Due to the high transmissibility of SARS-CoV, the discovery of novel vaccines is an urgent target to tackle this pathogen [312]. A traditional way to generate vaccine is the method of inactivated vaccines, which can be made with SARS-CoV-2 virions previously inactivated by physical or chemical treatments [313]. Currently, an attenuated vaccine that is generated by reducing the microbial virulence of a pathogen is known [314]. Biotechnology is mostly applied to generate vaccines in three ways: the separation of a pure antigen using a specific monoclonal antibody, the synthesis of an antigen with the assistance of a cloned gene, and the synthesis of peptides to be used as vaccines. Furthermore, producing active pharmaceutical chemical compounds in genetically modified organisms is a novel application of biotechnology achieved by molecular pharming or hairy root technologies.

### 10.1. Molecular Pharming: A Mature Technology to Produce Plant-Derived Pharmaceutical Products

Molecular pharming is one of the biotechnology tools that involves the application of various plant species for the production of recombinant proteins, which include enzymes, hormones, vaccines, and antibodies [315] (Figure 4).

Over the last few decades, a lot of efforts have been made to produce different bioactive compounds and proteins in high yield through plant genetic engineering approaches [316]. The molecular pharming platform has several advantages over other transgenic systems such as a low overall cost, a high scale-up capacity, safety, high product quality, and the ability of post-translational modifications [317]. In 2005, Pogrebnyak and colleagues [17] tried to develop a recombinant vaccine against SARS by expressing the N-terminal fragment of the SARS-CoV S protein (S1) in tomatoes and low-nicotine tobacco plants. The results of their experiment showed the high expression of recombinant S1 protein in transgenic lines. The production of plant-derived antigens significantly enhanced the levels of SARS-CoV-specific IgA after the oral ingestion of tomato fruits expressing the S1 protein. Sera of mice parenterally primed with tobacco-derived S1 protein revealed the presence of SARS-CoV-specific IgG as observed with Western blot and ELISA analysis. Two companies have announced their effort to produce plant-derived antibodies and vaccines against SARS-CoV-2. Soon after reporting the SARS-CoV-2 virus S protein, Medicago Inc. (www.medicago.com/en/pipeline/, the accessed date: 2 December 2020) announced VLP production in a transient gene expression system. Based on estimation, this company should be able to produce an attractive production capacity of VLP based vaccines (10 million doses). This amount of plant-derived vaccine can immunize approximately 10 million US adults who are at severe risk of critical illness or death if infected with SARSCoV-2. Furthermore, developing epitope-based vaccines is reported to reduce disease possibility. Kentucky Bioprocessing (www.kentuckybioprocessing.com, the accessed date: 2 December 2020), protalix (www.protalix.com, the accessed date: 2 December 2020), Greenovation Biopharmaceuticals (www.greenovation.com, the accessed date: 2 December 2020), Ventria (www.ventria.com, the accessed date: 2 December 2020), Nomad www.nomadbioscience.com, the accessed date: 2 December 2020), and iBio Inc. (www.ibioinc.com/pipeline, the accessed date: 2 December 2020) are other companies that are making efforts to produce a VLP vaccine. On the other hand, academic institutions such as the Infectious Disease Research Centre at Laval University and Medicago have joined hands together to develop plant-derived pipelines. Simultaneously, several institutes and universities from different countries including Thailand, Mexico, South Korea, South Africa, the UK, Germany, and the US are working on the production of plant-derived products through molecular farming technology against the Covid-19. It is believed that producing a stable plant-derived product through molecular farming takes a long time, and a major concern is that these products are not as approachable as expected in response to the pandemic. However, applying transient systems should be a reliable, fast, and efficient tool to overcome these barriers. Due to the high adaptability and potential of the SARS-CoV-2 virus, it seems that oral vaccines like plant molecular farming products are the most reliable weapons against the SARS-CoV-2 virus [318].

### 10.2. Hairy Root Culture: A Reliable Method of Producing Pharmaceutical Products

Hairy roots, which are induced by *Rhizobium rhizogenes* (syn. *Agrobacterium rhizogenes*), a Gram-negative bacterium, comprise an important biotechnological system to induce bioactive compounds from plants. Hairy root cultures employ organs instead of undifferentiated cells, thereby providing better yields than cell culture or natural plant roots [318,319]. Other advantages include biochemical and genetic stability, ease of preservation, and rapid growth on hormone-free media. Different *Agrobacterium* strains, which are divergent in terms of virulence, the growth rate of transformed cells, their morphology, and secondary metabolite production, are generally employed to induce hairy roots from explants in different plant species. It has been recognized that *Agrobacterium* strains have different abilities to promote the production of bioactive compounds in hairy root cultures [302,320,321]. This is mainly due to the differential expression and integration of T-DNA genes into the plant genome [302,321]. Therefore, the selection of an effective *Agrobacterium* strain for hairy root induction and secondary compound accumulation is noticeably dependent on plant species and must be empirically figured out [302,322] (Figure 5).

*Mentha spicata*, commonly known as spearmint or garden mint, is known to possess antiviral properties due to the presence of high amounts of phenolic compounds such as rosmarinic acid, chlorogenic acid, caffeic acid, lithospermic acid B, and cinnamic acid. The production of phenolic acids in *M. spicata* was compared among hairy roots induced from five different *A. rhizogenes* strains: ATCC15834, 9534, A13, A4, and R318. It was found that *A. rhizogenes* strain 9534 efficiently produced caffeic acid, lithospermic acid B, cinnamic acid, rosmarinic acid, and chlorogenic acid (106.76, 60.22, 44.02, 20.08, and 13.53 μg g^−1^ DW, respectively), but hairy root induction was effective with A13 and R318 [323,324,325,326].

It has been shown in many plant species that transformation by *A. rhizogenes* strains leads to the modification of metabolic pathways. As a result, transformed roots differ in their chemical profiles from normal roots, which shows that Ri T-DNA interferes with the biosynthesis of secondary metabolites [324,325,326].

The effects of different media (Schenk and Hildebrandt (SH), Woody plant medium (WPM), and Gamborg (B5)) and lighting conditions (light or dark) on biomass accumulation and secondary metabolite production in hairy root line (RC3) were examined in *Rhaponticum carthamoides* [326]. The WPM supported the highest biomass (93 g L^−1^ of the fresh weight after 35 days) under periodic light. Additionally, the higher production of caffeoylquinic acids and their derivatives was observed in hairy roots grown in the light as compared to untransformed roots. The biosynthesis of flavonoid glycosides such as quercetagetin, quercetin, luteolin, and patuletin hexoside from transformed roots was also found in light conditions [326]. Balasubramanian et al. [327] showed that time, number of co-cultivation days, acetosyringone concentration, media type, media strength, and sucrose concentration affect hairy root production for the improvement of quercetin content in *Raphanus sativus* (radish). Explants infected with an *A.*
*rhizogenes* MTCC 2364 suspension for 10 min and co-cultivated in a 1/2 MS medium containing acetosyringone (100 µM) for two days displayed a maximum percentage of hairy root induction (77.6%). Hairy roots were found to produce higher amounts of quercetin (114.8 mg g^−1^) compared to the auxin-induced roots of non-transformed radish. Similarly, higher amounts of phenolic compounds including pyrogallol, hesperidin, naringenin, and formononetin were observed in *Polygonum multiflorum* hairy roots compared to untransformed roots [328].

Because it has been found that both endogenous and exogenous factors affect resveratrol production in the hairy root culture of grapes and increase resveratrol production, this strategy could be useful [329]. Different features like the type of explants, seedling age, concentration of the bacterial inoculum, and inoculation time have been studied to improve the efficiency of hairy root formation and resveratrol production. Hairy roots induced by *A. rhizogenes* strain A4 from internodes of *Vitis vinifera* subsp. sylvesteris (W16) were found to produce the highest amount of resveratrol (31-fold higher than that of control root) [329]. In *Scutellaria baicalensis*, hairy roots grown in a full-strength MS medium produced a 2.5-fold higher amount of resveratrol than those grown in a half-strength B5 medium. The addition of auxin (indole acetic acid (IAA)) at 0.1 mg/L into the medium resulted in the highest accumulation of resveratrol [323,330].

The elicitation of hairy root cultures by biotic/abiotic stressors or chemicals for the exploitation of genetic engineering manipulation can significantly increase the amount of the desired secondary compound [331]. Different elicitors such as UV radiation, methyl jasmonate (MeJA), jasmonic acid, gibberellic acid, sodium acetate, salicylic acid, acetic acid, ammonium nitrate, chitosan, and cyclodextrins have been employed to stimulate the production of secondary metabolites in hairy roots [332]. Chitosan, methyl jasmonate, and yeast extract have been employed to enhance the glycyrrhizin contents in *Glycyrrhiza* species. Glycyrrhizin, an active component of licorice roots, is known to be effective against several viral diseases including HIV and SARS [104]. It has been reported that glycyrrhizin also inhibits the adsorption and penetration of the virus-early steps of the replicative cycle in addition to the inhibition of virus replication [104]. Elicitation with 100 μM MeJA enhanced glycyrrhizin content (5.7 times higher than the control) after five days of treatment in *G. inflata* hairy roots [333]. MeJA (at 100 μM concentration) was found to be the most effective elicitor for increasing glycyrrhizin production up to 108.9 ± 1.15 μg g^−1^ DW after five days of elicitation [334].

Elicitation with sodium acetate for 24 h resulted in the 60-fold induction and secretion of trans-resveratrol into a peanut hairy root culture medium [201]. The authors [335] studied the effect of different concentrations of abiotic elicitors including MeJA, sodium acetate, acetic acid, and ammonium nitrate. Their results showed that treatment of hairy roots with 3 mM acetic acid or 50 μM methyl jasmonate resulted in the highest or lowest amounts of hairy root biomass and resveratrol content, respectively.

## 11. Conclusions

The current review provides an overview of employing ethnomedicinal herbs to treat respiratory diseases. It has been shown that plant biodiversity can be a source of bioactive compounds of different natures, such as terpenes, stilbenes, coumarins, flavone glycosides, and alkaloids. The most common of these are lectins, quercetin, resveratrol, glycyrrhizin, and curcumin. It has been found that qualitative and quantitative biochemical specification can vary between representatives of the same family. Such a literature analysis could be used to choose an appropriate plant model for a specific region with specific needs. With the intervention of biotechnological tools such as hairy root transformation, it is possible to obtain compounds of interest in a higher amount. Plant-based production systems comprise another alternative for molecular farming technology. This review not only provides a reference point for the screening of plants against viral diseases but can also be useful for practical implications and applications.

## 12. Future Directions

The present review describes the potential of plant or plant-based compounds for treating antiviral diseases, especially emphasizing SARS-CoV. The urgency of this work is that new viruses may appear in the near future. There is a gap between information regarding the complete understanding of the mechanism of action of plant-based compounds and their execution as treatments of viral diseases. Both empirical and rational approaches are required to gain insight into the phytochemical evaluation and validation of plant-based compounds. Critical knowledge of genetic, molecular, and biochemical processes under in situ conditions could be helpful to better control the accumulation of natural products. Future studies may be targeted at understanding the mechanisms of action of complex plant extracts and isolated plant-derived compounds from different biosynthetic pathways. Molecular docking studies and in silico analyses of extracted compounds from different herbal plants including aloe vera and giloy could be beneficial for further investigations. By employing biotechnological tools like hairy root cultures and molecular pharming, it is quite possible to enhance the production of the compounds of interest. No doubt, careful preclinical and clinical procedures need to be followed before coming to the validation of plant-based drugs or vaccines against these viral diseases. This would help to pave the way and take important steps to combat this life-threatening and deadly disease.

## Figures and Tables

**Figure 1 molecules-26-03868-f001:**
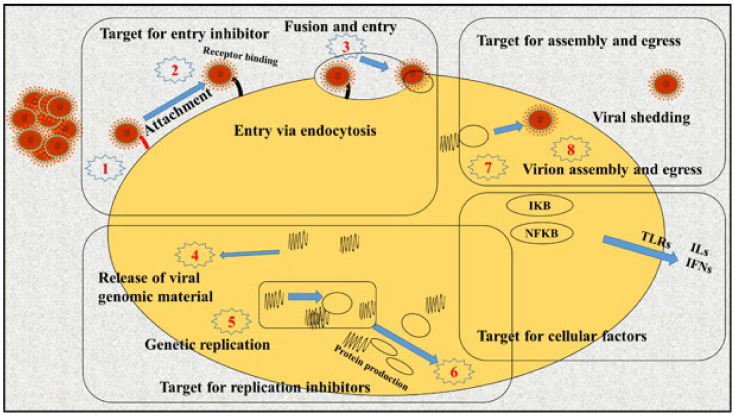
The most important viral replication chain includes virus attack to the host cell (Steps 1–3); entry using receptor bindings (Steps 1 and 4–6); and mRNA transcription/replication, and protein translation, and assembly and budding of progeny virus particles (Steps 1, 7, and 8). These steps are the most important goals for viral polymerase inhibition, replication (e.g., protease inhibitors, inhibitors of entry, and integrate inhibitors, among others), budding, and assembly [23].

**Figure 2 molecules-26-03868-f002:**
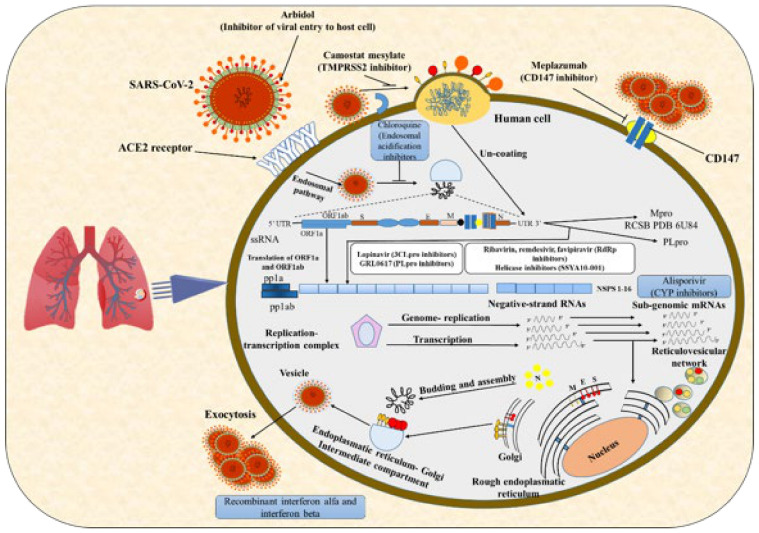
The replication cycle of SARS-CoV-2 and its inhibitors. The binding of the spike (S) protein to the host cell receptor initiates SARS-CoV-2 infection. SARS-CoV-2 has so far been linked to two cellular receptors: angiotensin-converting enzyme 2 (ACE2) and CD147. The cleavage of the S protein by the cell surface-associated enzyme occurs after receptor engagement. The viral genomic RNA is translated through ribosomal frameshifting to produce the polyproteins pp1a and pp1ab, which are co-translationally proteolytically processed into the 15 non-structural proteins (nsp1–nsp10 and nsp12–nsp16) that make up the replication-transcription complex (RTC). The RTC is involved in the replication of genomic RNA and the transcription of a series of nested subgenomic mRNAs that are essential for the expression of structural and accessory protein genes. New virions are formed by budding into the intracellular membranes of the ER–Golgi intermediate compartment membranes and then being released via exocytosis. In addition, blue denotes the extensive host-based therapy choices, and pink denotes specific viral-based treatment possibilities [30].

**Figure 3 molecules-26-03868-f003:**
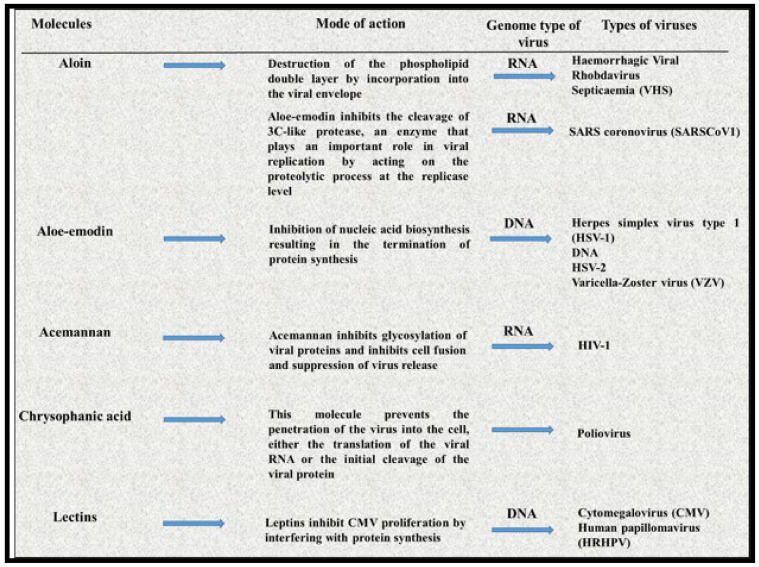
Antioxidant properties of *Aloe vera*.

**Figure 4 molecules-26-03868-f004:**
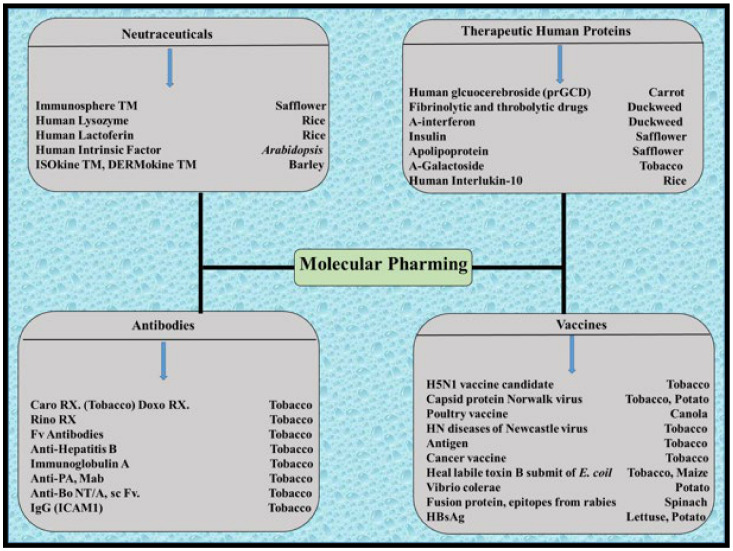
Plant molecular pharming products in different plants.

**Figure 5 molecules-26-03868-f005:**
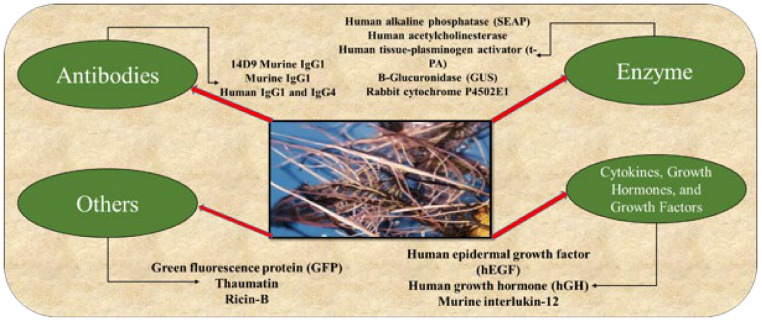
Hairy root cultures used to produce recombinant proteins. The final products of hairy root systems have been used to produce cytokines, vaccines, antibodies, enzymes, and other therapeutic proteins. These systems have various pros and cons, and these points should be considered when attempting to choose the best platforms.

**Table 1 molecules-26-03868-t001:** Example of biological active agents extracted from medicinal plants against SARS-CoV and other virus infections.

Plant Species or Plant Organ	Common Name	Active Against	Mode of Action	Compound(s) Isolated/Target	Reference
*Torreya nucifera*	Coniferous tree	SARS-CoV	3CL^pro^ inhibitor	Plant extract	[55]
*Thuja orientalis*	Oriental Arborvitae	SARS-CoV	Viral growth inhibitor	Plant extract	[35]
*Angelica keiskei*	Tomorrow’s leaf	SARS-CoV	3CL^pro^ inhibitor	Chalcones	[56]
*Radix glycyrrhizae*	Licorice root	SARS-CoV	Inhibits viral replication	Glycyrrhizin	[57]
*Dioscoreae Rhizoma*	Yam Rhizome	SARS-CoV	Viral growth inhibitor	Plant extract	[58]
*Psoralea corylifolia*	Babchi	SARS-CoV	PL^pro^ inhibitor	Plant extract	[59]
*Allium sativum*	Garlic	HCMV	Inhibits viral replication in earlier stages of viral cycle before viral DNA synthesis	Allitridin	[60]
*Myrica faya*	Fire tree	SARS-CoV	Helicase inhibitor	Myricetin	[61]
*Laggera* *pterodonta*	Curly Blumea	EV71	Inhibits viral RNA replication	Chrysosplenetin and penduletin	[62]
*Cassiae semen*	n/a	SARS-CoV	Viral growth inhibitor	Plant extract	[58]
*Cryptomeria japonica*	Sugi	SARS-CoV	Viral growth inhibitor	Hydroxy-deoxycryptojaponol	[63]
*Triterygium regelii*	Regel’s threewingnut	SARS-CoV	3CL^pro^ inhibitor	Celastrol Pristimerin Tingenone Iguesterin	[55]
*Gentianae Radix*	Gentian Root	SARS-CoV	Viral growth inhibitor	Plant extract	[58]
*Pterocarpus santalinus*	Red sandalwood	SARS-CoV	3CL^pro^ inhibitor	Savinin	[63]
*Betula pubescens*	Downy birch	SARS-CoV	3CL^pro^ inhibitor	Betulinic acid	[63]
*Fructus arctii*	Burdock	Flu	Inhibits viral replication	Arctigenin	[64]
*Galla chinensis*	Nutgall tree	SARS-CoV	Viral spike protein and Human ACE2 receptors inhibitor	Tetra-*O*-galloyl- β-d-glucose	[65]
*Saxifraga melanocentra* *Rhodiola kirilowii*	Saxifrages Golden root	HCV	Inhibits viral NS3 serine protease	Polyphenolic compounds (−)-Epicatechin-3-*O*-gallate, 3,30-digalloylproprodelphinidin B2, 3,30-digalloylprocyanidin B2, and (−)-epigallocatechin-3-*O*-gallate	[66,67]
*Paulownia* *tomentosa*	Princess tree	SARS-CoV	PL^pro^ inhibitor	Flavonoids	[67]
*Radix scutellariae* *Salvia miltiorrhiza* *Ranunculus sieboldii* *Ranunculus sceleratus* *Radix sophorae* *Radix bupleuri*	Huang Qin Red sage Buttercup Celery-leaved buttercup Shrubby sophora Thorowax Root	HBV	Inhibits viral DNA polymerase Inhibits viral replication Inhibits viral replication Inhibits viral replication Inhibits viral DNA replication and HBeAg production	Wogonin Protocatechuic aldehyde/Tanshinones Apigenin 40-*O*-a-rhamnopyranoside, apigenin 7-*O*-b-glucopyranosyl-40-*O*-a-rhamnopyranoside, tricin 7-*O*-b-glucopyranoside, tricin, isoscopoletin Apigenin 4′-*O*-α-rhamnopyranoside, apigenin 7-*O*-β-glucopyranosyl-4′-*O*-a-rhamnopyranoside, tricin 7-*O*-β-glucopyranoside, tricin, isoscopoletin Saikosaponin C	[16,68,69,70]
*Scutellaria lateriflora*	Blue skullcap	SARS-CoV	Helicase inhibitor	Scutellarein	[71]
*Loranthi ramus*	Mulberry	SARS-CoV	Viral growth inhibitor	Plant extract	[58]
*Stephania cepharantha*	n/a	SARS-CoV	Viral growth inhibitor	Biscoclaurine	[35]
*Cinnamomum cassia*	Chinese cassia	SARS-CoV	3CL^pro^ inhibitor	Plant extract	[71]
*Linum usitatissimum*	Flax	SARS-CoV	3CL^pro^ inhibitor	Herbacetin	[55]
*Alnus japonica*	East Asian alder	SARS-CoV	PL^pro^ inhibitor	Plant extract	[72]
*Laurus nobilis*	Bay laurel.	SARS-CoV	Viral growth inhibitor	Plant Extract	[35]
*Rhizoma coptidis* *Chrysanthemum* *Morifolium* *Vatica cinerea* *Aesculus chinensis* *Kadsura matsudai*	Huang Lian Florist’s daisy Resak Laut Chinese horse chestnut Kadsura	HIV/SARS-CoV	Inhibits viral DNA synthesis/3CL^pro^ inhibitor Inhibits viral integrase Inhibits viral replication Inhibits viral protease Inhibits viral replication	Berberine Apigenin-7-*O*-b-D-g-lucopyranoside Vaticinone (23E)-27-nor-3-hydroxycycloart-23-en-25-one Triterpenoid saponins Schizanrin B, C, D, and E	[73,74,75,76,77]
*Chamaecyparis obtuse* *Euphorbia jolkini* *Limonium sinense* *Ranunculus sieboldii* *Ranunculus sceleratus* *Limonium sinense*	Hinoki cypress Spurge Girard Buttercup Celery-leaved buttercup Girard	HSV	Inhibits HSV-1 ICP0, ICP4 expression, and as viral DNA synthesis Affects the late stage of HSV-2 Replication Inhibits viral replication Inhibits viral replication Inhibits viral replication	Yatein Putranjivain A Samarangenin B Protocatechuyl aldehyde Isodihydrosyringetin, (−)-epigallocatechin 3-*O*-gallate, samarangenin B, myricetin, myricetin 3-α-arhamnopyranoside, quercetin 3-*O*-α-rhamnopyranoside, (−)-epigallocatechin, gallic acid, N-trans-caffeoyltyramine, N-transferuloyltyramine	[16,78,79,80,81]
*Azadirachta indica*	Neem tree	Dengue virus	n/a	Leaf extract (Aqueous) inhibits DEN-2 in vivo	[82]
*Moringa oleifera*	Horseradish tree	HIV/Epstein-Barr virus (EBV)	n/a	Leaves used to inhibit viral replication/leaves and seeds inhibits activity against EBV activation	[83,84]
*Terminalia bellerica*	Myrobalan	HIV-1 Pseudo viruses	n/a	Plant extract against HIV-1	[85]
*Rheum palmatum*	Chinese rhubarb	SARS-CoV	Viral spike protein and	Emodin	[86]
*Avicennia marina*	Grey mangrove	Hepatitis B virus	Inhibits HBV antigen	n/a	[87]
*Litchi chinensis*	litchi	SARS-CoV	3CL^pro^ inhibitor	Flavonoids extract	[52]
*Multiflora Tuber*	Tuber fleeceflower	SARS-CoV	Viral spike protein and Human ACE2 receptors inhibitor	Emodin	[86]
*Canthium coromandelicum*	Alston	HIV	-	Leaf extract controls HIV infection	[82]
*Houttuynia cordata*	Fish mint	SARS-CoV	3CL^pro^ inhibitor and RNA-dependent RNA polymerase (RdRp) inhibitor	Plant extract	[88]
*Veronica linariifolia*	Speedwell	SARS-CoV	Viral growth inhibitor	Luteolin	[62]
*Carissa edulis*	Conkerberry	Herpes simplex virus	Exhibits anti-HSV-1 and -2 properties in vitro and in vivo strongly	n/a	[89]
*Nicotiana benthamiana*	Benth	SARS-CoV	Viral growth inhibitor	NICTABA Lectin	[90]
*Urtica dioica*	Common nettle	SARS-CoV	Viral spike protein inhibitor	*Urtica dioica* agglutinin	[91]
*Phyllanthus amarus*	Indian gooseberry	Human immunodeficiency virus/hepatitis B virus	Inhibits HIV replication	Plant extract had lost HBV antigen surface	[92]
*Guazuma ulmifolia Lam*	West Indian elm	Polio virus	Extracts inhibits polio replications	n/a	[93]
*Achyranthus aspera*	Chaff-flower	Herpes simplex virus	Inhibits earlier stages of HSV multiplications	n/a	[94]
*Camelliasinensis*	Tea tree	SARS-CoV	3CL^pro^ inhibitor	Tannic acid/3-isotheaflavin-3-gallate	[95]
*Sesbania grandiflora*	Vegetable hummingbird	Herpes simplex virus	n/a	Extract possesses strong antiviral activity against HSV	[96]
*Ficus religiosa*	Bo tree	Human rhino virus (HRV) and Respiratory syncytial virus (RSV)	n/a	Bark extract endowed with antivirus activity against HRV and RSV	[97]
*Hippophae rhamnoides*		Dengue virus	Significant anti-dengue activity	Leaf extract	[98]
*Glycine max* *(black)*	Soybean	Human adenovirus (type 1)	Inhibits human ADV-1 in dose-dependent manner	n/a	[99]
*Acacia nilotica*	Gum arabic tree	HIV-PR	Inhibition	n/a	[100]
*Allium sativum*	Garlic	SARS	Proteolytic and hemagglutinating activity and viral replication	n/a	[101]
*Andrographis* *paniculata* *Boerhaavia diffusa*	Green chireta Punarnava	SARS-COV and likely SARS-CoV-2	Suppression Inhibition	NLRP3, capase-1, and IL-1β ACE	[31,102,103]
*Clerodendrum inerme*	The glory bower	SARS-CoV-2	Inactivation	Ribosome	[104]
*Clitoria ternatea*	Butterfly pea	n/a	Metalloproteinase inhibitor	ADAM17	[105]
*Coriandrum sativum*	Coriander	n/a	Inhibition	ACE	[106]
*Cynara scolymus* *Cassia occidentalis*	Scolymus	n/a	Inhibition	ACE	[102,103]
*Embelia ribes*	White-flowered Embelia	n/a	Inhibition	ACE	[102,103]
*Eugenia jambolana*	Black Plum	n/a	Inhibition	Protease	[49]
*Euphorbia granulata*	Asthma-plant	HIV-1 PR	Inhibition	-	[100]
*Glycyrrhiza glabra* *Gymnema sylvestre*	Licorice Gurmar	SARS; HIV-1	Inhibition of viral replication; modulation of membrane fluidity Inhibition of viral DNA synthesis	Glycyrrhizin	[57,107]
*Hyoscyamus niger*	black henbane	n/a	Inhibition and Bronchodilator	Ca^2+^	[108]
*Ocimum* *kilimandscharicum*	Camphor basil	HIV-1	Inhibition	n/a	[51]
*Ocimum sanctum*	Holy basil	HIV-1	Inhibition	n/a	[50]
*Punica granatum*	Pomegranate	Human herpes virus-3	Inhibition	ACE/Phytochemical extract exhibits potential antiviral activity	[102,103]
*Salacia oblonga*	Oblong leaf salacia	n/a	Suppression	Angiotensin II and AT1	[109]
*Sambucus ebulus*	Danewort	Enveloped virus	Inhibition	n/a	[110]
*Solanum nigrum*	European black nightshade	HIV-1	n/a	n/a	[52]
*Sphaeranthus indicus*	East Indian globe	Mouse corona virus and Herpes virus	Inhibition	n/a	[111,112]
*Strobilanthes callosa*	Plietesials	HCoV-NL63	Blocking	n/a	[107,113]
*Strobilanthes cusia*	Kuntze	HCoV-NL63	Blocking	n/a	[113]
*Vitex negundo*	Five-leaved chaste tree	HIV-1	Inhibition	n/a	[52,53,113,114]
*Emblica officinalis*	Emblic	Influenza A virus	Prevention of virus adsorption and suppression of virus release	Pentagalloyl glucose	[114]
*Vitex trifolia*	Simpleleaf Chastetree	SARS-COV	Reduction	n/a	[115]

**Table 4 molecules-26-03868-t004:** List of commonly used medicinal herbs involved in treatment of respiratory disorders.

Scintific Name	Common Names	Mode of Action	References
*Tinospora cordifolia*	Heart-leaved moonseed	Chronic fever	[257,258]
*Andrograhis paniculata*	Creat or green chireta	Fever and cold	
*Cydonia oblonga*	Quince	Antioxidant, immune-modulatory, anti-allergic, smooth muscle relaxant, and anti-influenza activity	
*Zizyphus jujube*	Jujube		
*Cordia myxa*	lasura		
*Agastya haritaki*	*Agastya Rasayana*	Upper respiratory infections	
*Anu Thailam*	-	Respiratory infections Fever	
*Adathoda siddha*	Malabar nut	Fever	
*Bryonia alba*	White bryony	Reduce lung inflammation	
*Rhus toxicodendron*	Eastern Poison Oak	Viral infections	
*Atropa* *belladonna*	Deadly nightshade	Asthma and chronic lung diseases	
*Bignonia* *sempervirens*	Yellow jessamine	Asthma	
*Eupatorium perfoliatum*	Agueweed	Respiratory symptoms	
*Kabasura Kudineer*	-	Fever, cough, sore throat, shortness of breath	
*Abutilon indicum*	Mallow	Immunomodulatory function	[107,259]
*Withania somnifera*	Ashwagandha	General tonic and to boost immunity/against herpes simplex virus	[260]
*Hydrastis canadensis*	Goldenseal root	Reduces plasma TNF-α, INF-γ, and NO levels; inhibits the T helper-type 2 cytokine profile.	[261]
*Acalypha indica*	Indian Acalypha	Anthelmintic	[262]
*Achyranthes aspera*	Prickly chaff flower	Anti-viral activity	[263]
*Momordica charantia*	bitter melon	Inhibits the release of TNF-α, NO and proliferation of spleen cells induced by PHA and Con A.	[261]
*Adhatoda vasica*	Adulsa	Anti-asthmatic, anti-allergic and anti-tussive activity	[264]
*Nigella sativa*	Black cumin	Reduces the pancreatic ductal adenocarcinoma cell (PDA) synthesis of monocyte chemoattractant protein-1 (MCP-1), TNF- α, IL-1β, and cyclooxygenase (COX)-2, as well as inhibiting the polymorphonuclear leukocyte functions.	[261]
*Alangium salvifolium*	Sage-leaved alangium	Anti-rheumatoid	[265]
*Urtica dioica*	Common Nettle	Reduction of TNF-α and other inflammatory cytokines	[261]
*Cassia alata*	Candle Bush	Anti-helmintic activity	[107,266]
*Cassia fistula*	Golden Shower	Antibacterial activity	[267]
*Cayratia pedata*	Birdfoot Grape	Anti-inflammatory activity	[268]
*Chloroxylon swietenia*	East Indian satinwood	Anti-helmintic activity	[269]
*Clitoria ternatea linn.*	Asian pigeonwings	Anti-oxidant activity	[270]
*Eugenia singampattiana*	Jungle Guava	Anti-inflammatory activity	[107,268]
*Hippophae rhamnoides*	Sea-buckthorn	Eliminating phlegm, stopping coughing, improving digestion, and treating lung diseases	[271]
*Indigofera tinctoria*	True Indigo	Immunomodulatory function	[107,272]
*Justicia adhatoda*	Malabar Nut	Anti-oxidant and anti-mutagenic activity; hepatoprotective activity	[1,236,273,274,275]
*Leucas aspera*	Thumbai	Hepatoprotective activity, acaricidal properties	[107,276,277]
*Mucuna pruriens* *Xanthium strumanium*	Monkey Tamarind Common Cocklebur	Antibacterial drugs against pneumonia	[278]
*Pergularia daemia*	Pergularia	Hepatoprotective effect	[107,279]
*Piper longum*	Long pepper	Anti-pneumonia drug	[280]
*Salacia reticulata*	Meharimula	Anti-inflammatory activity	[268]
*Santalum album*	White Sandalwood	Anti-inflammation of the lungs, blood, and pus in the sputum	[271]
*Solanum tornum*	Turkey Berry	Anti-pneumonia drug	[281]
*Solanum suratens*	Surattense Nightshade	Anti-viral activity	[263]
*Solanum xanthocarpum*	Yellow-fruit nightshade	Anti-asthmatic	[264,271]
*Strychnos minor*	Snakewood	Anti-inflammatory activity	[268]
*Strychnosnux vomica*	Strychnine tree	Anti-inflammatory activity	[268]
*Syzygium aromaticum*	Caryophyllus	Hepatoprotective properties	[282]
*Tinospora cordifolia*	Moonseed	Anti-pneumonia drug	[283]
*Trichopus zeylanicus*	Arogyapacha	Anti-oxidant and anti-fatigue activity	[284]
*Tylophora indica*	Indian Ipecac	Anti-asthmatic	[264]
*Verbascum thapsus*	Great mullein	Enhancing peroxidase, phenolics, and antioxidant activity	[1]
*Vitex altissima*	Peacock Chaste Tree	Acts against acute inflammation	[285]
*Vitex trifolia*	Hand-of-Mary	Tracheospasmolytic activity	[107,257]
*Wrightia tinctoria*	Pala Indigo	Anti-inflammatory activity	[268]
*Yuthog’s Bamboo*	Bamboo	Reduction of pain and anti-inflammation of the lungs and respiratory tract	[271]
*Zingiber officinale*	Canton Ginger	Modulation of macrophage functions, phagocytic properties, anti-viral activity, and bronchial infections	[1,286,287]
*Garcinia Kola*	Bitter Kola	Anti-bacterial drug against respiratory pathogens	[288]
*Cymbopogon citratus*	West Indian lemon grass	Anti-viral infections	[289]
*Camellia sinensis*	Tea plant	Anti-infective activity	[290]
*Achillea mellefolium*	Yarrow	Protects upper respiratory tract from viral infections	[291]
*Apium graveolens*	Celery	Anti-bacterial and anti-viral agent	[292]
*Borassus flabellifer*	Palmyra Palm	Protects from pulmonary infections; anti-bacterial and anti-viral activity	[293]
*Caesalpinia bonduc*	Grey Nicker	Treatment for asthma (anti-bacterial and anti-viral agent)	[294]
*Calotropis gigantea*	Crown flower	Anti-bacterial and anti-viral agent	[293]
*Crocus sativus*	Saffron Crocus	Treatment for asthma and cough	[295]
*Euphorbia hirta*	Asthma-Plant	Anti-bacterial and anti-viral agent	[291]
*Piper nigrum*	Black pepper	Anti-viral agent	[296]
*Strychnos potatorum*	Clearing-Nut Tree	Treatment for bronchitis	[297]
*Terminalia bellirica Roxb*	Beleric Myrobalan	Effective for asthma	[298]
*Tylophora indica Merrill*	Antamul	Treatment for bronchitis and asthma	[299]
*Tussilago farfara*	Coltsfoot	Treatment for cough and asthma	[300]
*Thymus linearis*	Himalayan Wild Thyme	Anti-viral activity	[290]
*Senecio chrysanthemoides*	Senecio	Treatment for lung diseases	[301]
*Portulaca oleracea*	Duckweed	Anti-inflammatory and anti-asthma properties	[294]
*Papaver somniferum*	Opium Poppy	Treatment for respiratory diseases	[293]
*Morus laevigata Wall. ex Brandis*	White Mulberry	Treatment for cough	[302]
*Ephedra gerardiana Wall. ex Stapf*	Gerard’s Jointfir	Treatment for cough and asthma	[303]
*Geranium wallichianum*	Sylvia’s Surprise	Treatment for cough	[292]
*Micromeria biflora*	Lemon Scented Thyme	Treatment for cough	[295]
*Picrorhiza kurroa Royle ex. Benth*	Kutki	Treatment for asthma and bronchitis	[289]
*Primula denticulata Sm.*	Drumstick Primula	Treatment for cough and bronchitis	[304]

**Table 5 molecules-26-03868-t005:** In silico analysis of phenolic compounds extracted form medicinal plants.

Plant Parts	Scientific Name	The Major Phenolic Compound	Formula	Active Sites/Binding Residue/H-Bond Length (Å)
Leaves of harsingar	*Nyctanthes arbortristis*	Nictoflorin	C_27_H_30_O_15_	N-H--O/GLY-143/2.311
		Astragalin	C_21_H_20_O_11_	O--H/PHE-140/2.197
		Lupeol	C_25_H_26_O_4_	N-H--O/THR-26/2.027
Giloy	*Tinospora cordifolia*	Berberine Sitosterol	C_28_H_18_NO_4_ C_29_H_50_O	N-H--O/GLY-143/2.540 N-H--O/GLY-143/2.577 O--H/PHE-166/2.080
*Aloe vera* leaves	*Aloe barbadensis*	Aloenin Aloesin	C_19_H_22_O_10_ C_19_H_22_O_9_	O--H/PHE-140/2.151 N-H--O/GLY-143/2.016 N-H--O/GLU-166/2.297
The dried ground rhizome of the turmeric	*Curcuma longa*	Curcumin	C_21_H_20_O_6_	N-H--O/GLY-143/2.243
The oil of neem	*Azadirachta indica*	Nimbin	C_30_H_36_O_9_	N-H--O/GLY-143/2.161
Steroidal constituents of ashwagandha		Withanolide Withaferin A	C_28_H_38_O_6_ C_28_H_38_O_6_	O--H/GLU-166/1.991 N-H--O/GLU-166/2.110 N-H--O/GLY-143/2.577
Constituents of pungent ketones, which result in the strong aroma of ginger	*Zingiber officinale*	Gingerol Shogaol	C_17_H_26_O_4_ C_17_H_24_O_3_	O--H/THR-190/2.026 N-H--O/GLY-143/2.289 N-H--O/THR-26/2.398 O--H/THR-24/2.345
Red onion	*Allium cepa*	Quercetin	C_15_H_10_O_7_	O--H/THR-26/1.936
Tulsi leaves	*Ocimum sanctum*	Ursolic acid Apigenin	C_30_H_48_O_3_ C_15_H_10_O_5_	N-H--O/GLY-143/2.330 O--H/THR-26/1.994
Cannabis extracts	*Cannabis sativa*	Cannabidiol	C_21_H_30_O_2_	N-H--O/GLY-143/2.325
Isolated from the plants of both the black and white pepper grains	*Piper nigrum*	Piperine	C_17_H_19_NO_3_	N-H--O/THR-26/2.529

## Data Availability

Not applicable.

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
