# Peer review of "A Brief Overview of Potential Treatments for Viral Diseases Using Natural Plant Compounds: The Case of SARS-Cov"

_molecules, 2021, doi:10.3390/molecules26133868_

Round 1

Reviewer 1 Report

The review manuscript reported by R  Abiri et al, to “A Brief Overview of Potential Treatments for Viral Diseases using Natural Plant Compounds: The Case of SARS-Cov”. The scope of the study will be helpful to enhance the graph of working against SARS-CoV. I believe this study based on SARS-CoV2 has the merit to be published in Molecule’s journal. However, I have noted some problems while reviewing this study, which need to be addressed.

  1. In the Abstract section should be re-written to talk about latest litrature.
  2. The background is too short for the reader to understand easily please add some recent literature.
  3. Please also add some new literature.
  4. The discussion is too long please insuring it cover some aspects of methodology and results it will be helpful for readers.
  5. The review needs some figures to be included.
  6. Do you think Nigella sativa can be used for the treat of SARS-CoV2?
  7. English and grammatical error must be improved.
  8. Need to cite these papers have a look 1. doi.org/10.1080/07391102.2020.1769733, 2. https://doi.org/10.1007/s12539-020-00381-9, 3. https://doi.org/10.1016/j.csbj.2020.08.006, 4.https://doi.org/10.1007/s00203-020-01998-6

Author Response

Reviewer 1

The review manuscript reported by R  Abiri et al, to “A Brief Overview of Potential Treatments for Viral Diseases using Natural Plant Compounds: The Case of SARS-Cov”. The scope of the study will be helpful to enhance the graph of working against SARS-CoV. I believe this study based on SARS-CoV2 has the merit to be published in Molecule’s journal. However, I have noted some problems while reviewing this study, which need to be addressed.

  1. In the Abstract section should be re-written to talk about latest litrature.

Reply: Abstract is re-written as suggested.

  1. The background is too short for the reader to understand easily please add some recent literature.

Reply: Recent literature is included.

  1. Please also add some new literature.

Reply: Included as suggested

  1. The discussion is too long please insuring it cover some aspects of methodology and results it will be helpful for readers.

Reply: It is amended as suggested.

  1. The review needs some figures to be included.

Reply: Figures have been re-framed.

  1. Do you think Nigella sativa can be used for the treat of SARS-CoV2?

Reply: Yes, it is reported in literature for its antiviral properties.

  1. English and grammatical error must be improved.

Reply: English language and grammar is improved by the native English speaker.

  1. Need to cite these papers have a look 1. doi.org/10.1080/07391102.2020.1769733, 2. https://doi.org/10.1007/s12539-020-00381-9, 3. https://doi.org/10.1016/j.csbj.2020.08.006, 4.https://doi.org/10.1007/s00203-020-01998-6
  2. Reply: Included

Reviewer 2 Report

This manuscript reviewed the antiviral and coronavirus potential of phytochemicals extracted from plants. Everything seem to be properly done with adequate experts. The paper is totally well-written and would be much of interest to the readers. After reading the manuscript over, my evaluation is that this paper would be appropriate for MOLECULES.

This manuscript, although it talks about a current and particularly critical issue (Sars-cov-2), brings few novelties about what has been recently published and revised. Thus, it does not mention the new literary reviews that address the same theme. However, the authors propose a brief review, and its can be improved by:   Inserting some new references and revisions, which talks about phytochemicals and sars-cov-2

I recommend improving the figures resolutions, and its fonts to better visualization

Author Response

Reviewer 2

This manuscript reviewed the antiviral and coronavirus potential of phytochemicals extracted from plants. Everything seem to be properly done with adequate experts. The paper is totally well-written and would be much of interest to the readers. After reading the manuscript over, my evaluation is that this paper would be appropriate for MOLECULES.

This manuscript, although it talks about a current and particularly critical issue (Sars-cov-2), brings few novelties about what has been recently published and revised. Thus, it does not mention the new literary reviews that address the same theme. However, the authors propose a brief review, and its can be improved by:   Inserting some new references and revisions, which talks about phytochemicals and sars-cov-2

I recommend improving the figures resolutions, and its fonts to better visualization

Reply: Recent literature is included as suggested and figures with high resolutions are included.

Reviewer 3 Report

The authors report an overview on the potential antiviral treatments using medicinal plants. The topic is interesting and the review overall well described.

However, I have some concerns outlined below:

  • the authors mentioned Sars-Cov. Wouldn't it be more appropriate to define? Cov 2 for example? This is also the case in Table 1.
  • Suddenly in Table 1 other viruses, including HIV, HBV and HCV are reported. There is no mention of these viruses in the previous text. 
  • Why some sentences in paragraph 7 are in italics?
  • The authors then moved on exploring the effect of isolated compounds against respiratory viruses. I believe the review needs to be restructured concentrating on a group/family of viruses.
  • I am not sure the vaccine production section is relevant.

Finally, the authors should correct several linguistic irregularities. 

Author Response

Reviewer 3

The authors report an overview on the potential antiviral treatments using medicinal plants. The topic is interesting and the review overall well described.

However, I have some concerns outlined below:

  • the authors mentioned Sars-Cov. Wouldn't it be more appropriate to define? Cov 2 for example? This is also the case in Table 1.

Reply: Done as suggested

  • Suddenly in Table 1 other viruses, including HIV, HBV and HCV are reported. There is no mention of these viruses in the previous text. 

Reply: Included in the text

  • Why some sentences in paragraph 7 are in italics?

Reply: The said sentences have been corrected.

  • The authors then moved on exploring the effect of isolated compounds against respiratory viruses. I believe the review needs to be restructured concentrating on a group/family of viruses.

Reply: Done as suggested

  • I am not sure the vaccine production section is relevant.

Reply: It may give an overall idea to the readers form other fields. However, if reviewer thinks this part is not necessary then we have no objection to delete this section.

Finally, the authors should correct several linguistic irregularities. 

Reply: Corrected as suggested

Round 2

Reviewer 1 Report

It should be accepted as is

Reviewer 3 Report

The authors have revised the manuscript accordingly.